# Utilization of the *Viscum* Species for Diet and Medicinal Purposes in Ruminants: A Review

**DOI:** 10.3390/ani12192569

**Published:** 2022-09-26

**Authors:** Onke Hawu, Khuliso Emmanuel Ravhuhali, Mutshidzi Given Musekwa, Nkosomzi Sipango, Humbelani Silas Mudau, Kwena Hilda Mokoboki, Bethwell Moyo

**Affiliations:** 1Department of Animal Science, School of Agricultural Sciences, Faculty of Natural and Agricultural Sciences, North-West University, Mmabatho 2735, South Africa; 2Food Security and Safety Niche Area, Faculty of Natural and Agricultural Sciences, North-West University, Mmabatho 2735, South Africa; 3Department of Animal Production, Fort Cox Agriculture and Forestry Training Institute, Middledrift 5685, South Africa

**Keywords:** nutritive value, semi-arid, protein, host, parasitic

## Abstract

**Simple Summary:**

True mistletoe (of the *Viscum* species) is a semi-parasitic, perennial browse species that is found attached to its host—a shrub or a tree. It has important pharmaceutical and chemical properties that allow it to be used for a variety of purposes, including livestock production. Mistletoes are cheap and are a readily available source of minerals and protein for livestock, especially during the dry season. They grow primarily on the outer branches of the tree crown; however, they also frequently grow directly on the tree trunk in order to consume its nutrients and water, thereby affecting their host’s quality and development. Many countries are expected to discover and explore the potential of *Viscum* spp. and their management tools, which we investigate through this review.

**Abstract:**

A cost-effective, alternative protein and mineral source such as the *Viscum* species can be key to livestock production. *Viscum* spp. are used as feed in many semi-arid and arid regions globally, particularly during feed shortages. The species’ feeding value, and their pharmaceutical attributes, have been recognized worldwide, albeit with variation in nutritive value from one host to another. The antinutritional factors found in *Viscum* spp. may benefit livestock when consumed in moderation due to their immunomodulatory, proapoptotic, and antimicrobial properties. The *Vachellia* species are known to be the common hosts for *Viscum* spp. Further, even though *Viscum* spp. inhibit host tree development by reducing carbon absorption and the host tree’s carbohydrates, the efforts to regulate their infestation should not result in the plant’s total eradication due to the benefits to livestock (as well as in fodder and medicine). This review will help to improve understanding of *Viscum* species control measures, while also increasing the productivity of ruminants.

## 1. Introduction

Finding inexpensive alternative protein sources such as the *Viscum* species is necessary since livestock productivity has continued to be severely constrained by the cost of livestock feed. True mistletoe (of the genus *Viscum*) is a semi-parasitic, perennial browse species that attaches to its host, shrubs or various tree species [1,2]. It is mainly dispersed by frugivorous birds from one host to another [1]. It has fodder value, as well as anthelmintic and therapeutic properties with evergreen leaves [2,3]. The genus *Viscum* contains many species that are primarily found in America, Africa, Asia, and Europe [4]. The *Viscum* spp. found in southern Africa include *V. verrucosum, V. rotundifolium, V. anceps, V. songimveloensis*, and *V. combreticola* [5,6,7]. They are fodder resources for ruminants, especially during dry periods when good quality forage is scarce [8]. Öztürk et al. [3] highlighted that the *Viscum* species extract nutrients and water from their host; hence, they are a rich fodder resource for ruminants.

True mistletoes are ingested and preferred by livestock without any reported digestive orders [9]. Even though their ecological importance for birds, medicinal properties, and fodder value for livestock are known, they are still regularly removed from orchards and rangelands/forests due to their detrimental effect on the host plant [10]. This research aims to help improve understanding of *Viscum* species control measures, while also increasing the productivity of ruminants. This means that developing mitigation strategies to minimize its spread should take into account a more balanced understanding that incorporates knowledge of its nutritive value as a source of protein, as well as its negative impact on rangelands. In this paper, we reviewed the primary uses of *Viscum* spp. in the livestock industry as well as in other human endeavors.

## 2. Description of the *Viscum* Species

True mistletoe (*Viscum* spp.) is an evergreen hemiparasitic plant that inhabits trees. Yellowish flowers, small yellowish green leaves, and waxy, white berries characterize this parasitic plant (Figure 1). Some of the species have leaves while some do not have leaves (Table 1). For example, *V. album*, when on the branch of a host tree, will grow as much as 60–90 cm long with a drooping yellowish evergreen shrub. It has densely packed forking branches that are 5 cm long, leathery, oval- to lance-shaped leaves that are placed in pairs on branches. The bisexual, or unisexual, blooms are arranged in tight spikes and have consistent symmetry [11]. However, some *Viscum* spp. have smooth, round, green stems that are covered in sessile, yellowish blooms in tiny clusters (Figure 2). The flowers of the Viscaceae family are narrow, tubular, dioecious, with (or without) a corolla, and thus pollinated by insects and the wind [12].

## 3. Adaptation of the Species

*Viscum* spp. grow on the branches of various tree species. They extract nutrients and water from the host plant for their survival [22]. Although their leaves may photosynthesize, they do so at a slower rate than their hosts [23]. Ahmad et al. [24] highlighted that they contain a functionally low amount of chlorophyll, and their low capability for photosynthesis explains their capability to adapt to dry conditions. They can survive in semi-arid regions, deserts, temperate woodlands, and semi-tropic wetlands [25]. It has been suggested that true mistletoes selectively parasitize host species that are high in nitrogen since nitrogen is frequently a limiting resource for plants [26]. Moreover, in South Africa, the genus *Vachellia* are the most important hosts of *Viscum* spp. Clark et al. [27] highlighted that there are just four *Viscum* species in South Africa that are unique or specific to a single host, which is a relatively low number.

## 4. Negative Impact and Control of the *Viscum* Species

It has been extensively researched for years how common *Viscum* spp. affect woody species, particularly in rangelands and in plantations. Mistletoe inhibits host tree development by reducing carbon absorption and host tree carbohydrates, all of which have an impact on the quality and quantity of woody species produced and the soil’s nutrient cycle [28]. Within its current range, mistletoe abundance has been growing, and the intensification of climatic stress in the form of protracted droughts has increased the rate of tree mortality in mistletoe-infected woody species, thus altering the dynamics of the community [29]. Moreover, true mistletoe spp. induce nutrient and water stress, which, in turn, changes the phyto-hormone profile, as well the defense mechanism of the host plant and causes affected trees to be more susceptible to insect attacks [30]. To overcome such problems, mistletoe spp. infestations should be controlled or managed in the rangelands.

*Viscum* spp. can be controlled using mechanical, chemical, or biological means. The single most successful approach to eradicate mistletoe in rangelands or forests is mechanical removal of mistletoe by clipping infected branches; however, this requires a large amount of labor and finances [28]. The use of chemicals as a control measure has been documented. Further, injecting a chemical into the trunk of a plant with mistletoe has been proposed [31]. However, this method does not address the root of the infestation and entails the possibility that the dosage will either fail to eradicate the mistletoe or harm the host plant. Livestock browse preferably on mistletoes when available; this, therefore, suggests that livestock can be used as biological agents to control the spread of mistletoe spp. However, it is unknown whether livestock have a comparable preference for mistletoes on plant hosts.

## 5. Crude Protein and Fiber Fraction of *Viscum* Species

The high prices of livestock’s more conventional feeds make *Viscum* spp. a nutritionally suitable feed for ruminants during particularly dry periods. Grasses during this period normally deteriorate and lose their nutritive value. The nutritive value of *Viscum* spp. usually varies from one host to another due to link-specific nutrient transfer characteristics [32]. Previous studies have reported that *Viscum* spp. have a crude protein (CP) content of more than 80 g/kg DM, which is considered to be enough for rumen microbes in growing ruminants (cattle, sheep, and goats) [2,33]. This further highlights the importance of *Viscum* spp. during the dry season, as they address protein deficiencies when the CP content of grasses is between 20 and 60 g/kg DM. Hawu et al. [34] highlighted that low CP content usually decreases feed intake, and adversely affects ruminant growth and productivity.

The fiber content of forage is one of the most vital parameters to consider as this will affect both feed intake and digestibility for ruminants. *Viscum* spp. contain relatively low fiber concentrations, as shown in Table 2; this is due to their low photosynthesis capacity. *Viscum* spp. may not produce some more complex carbon materials such as fiber, which are, however, produced by other woody browse species [35]. Consequently, *Viscum* spp. do not have high acid detergent fiber, neutral detergent fiber, or acid detergent lignin content, thus making them highly digestible. Therefore, the low fiber content in *Viscum* spp. does not constrain the use of *Viscum* spp. as a fodder for ruminants that are adept at utilizing forages that are high in fiber.

## 6. Potential of *Viscum* Species as a Source of Minerals for Ruminants

Minerals play an important role in the metabolic functions of livestock. These functions assist with supporting growth, development, immune function, and the reproductive performance of livestock [39,40]. *Viscum* spp. are known as a source of minerals such as phosphorous (P), iron (Fe), calcium (Ca), magnesium (Mg), zinc (Zn), copper (Cu), and other minerals that are required for ruminants’ wellbeing (Table 3). Umucalılar et al. [41] reported average Ca (13 g/kg), P (3 g/kg DM), Fe (110 g/kg DM), Cu (10 g/kg DM), and Zn (41 g/kg DM) in *V. album* from different plant hosts.

Numerous physiological processes depend on calcium. Calcium (Ca) plays an important role in blood clotting, membrane permeability, nerve conduction, muscle contraction, enzyme activity, and hormone secretion [42,43]. The concentration level of Ca in *Viscum* spp. is higher than the 5.8 g/kg required by growing calves [44]. However, there may be a need to reduce the Ca concentration level in ruminant diets that contain *Viscum* spp. in order to avoid toxicity. Phosphorus is an essential component of adenosine triphosphate (ATP) and nucleic acid, it is also important for the formation of teeth and bones [45]. The concentration level of P in *Viscum* spp. is equivalent to the 2 g/kg that is required by lactating cows [44]. Iron is required for the synthesis of hemoglobin and myoglobin, as well as several other enzymes that aid in the formation of ATP via the electron transport chain [46]. Hill and Shannon [47] highlighted that Zn plays a variety of roles in immunity and disease resistance. Moreover, Zn is essential for growth and cell division, where it is required for protein and DNA synthesis, insulin activity, ovary and testis metabolism, and liver function [48,49,50]. Copper performs a physiological role in cellular respiration, bone development, heart health (functions), the formation of connective tissue, the myelination of the spinal cord, and in keratinization and pigmentation processes [51,52]. The concentration level of Cu in *V. verrucosum* is equivalent to the 0.01–0.02 g/kg required by growing lambs [53]. These mineral values, as mentioned above, suggest that *Viscum* spp. can be fed to ruminants without mineral supplementation since these values are higher than the minimum mineral requirement [54,55].

**Table 3 animals-12-02569-t003:** Mineral content (g/kg DM) of *Viscum* ssp.

Species	Ca	K	P	Mg	Na	Zn	Cu	S	Fe	Mn	References
*V. album*	13	25	3	32.57		2	1.1	14.4	29.2	99	[41,56,57]
*V. verrucosum*	76	97	2	7	1.2	0.02	0.03		0.44	0.05	[2]

## 7. Antinutritional Factors Associated with *Viscum* ssp.

Plants use phytochemicals as a defense mechanism against diseases and other external threats [58]. There is increasing interest in studying the bioactivity and the antinutritional factors (ANFs) (phytochemicals) of *Viscum* spp. To clarify, antinutritional factors are plant components that have the potential to negatively impact livestock productivity. Several authors have reported the presence of ANFs in *Viscum* spp., as shown in Table 4. García-García et al. [57] reported that *Viscum* spp. contain ANFs such as tannins, saponins, alkaloids, and flavonoids; further, *Viscum* spp. depend on the host they grow on. In contrast, it has been discovered that some ANFs might benefit livestock when consumed in moderation. Wang et al. [59] stressed that flavonoids have various bioactive effects, such as cardio protective, anti-inflammatory, and antiviral. Saponin has a number of biological effects on livestock, such as hemolysis of erythrocytes, a decrease in blot (in ruminants), a reduction in the activity of smooth muscles, an inhibition of enzymes, a reduction in nutrient absorption, and an alteration in cell wall permeability, and thus produces some poisonous effects when ingested [60,61]. High tannin concentrations in ruminants are known to reduce palatability, feed intake, and degradability [34], while low tannin concentrations are known to have health benefits such as antiviral and antibacterial effects [62]. In relation to greenhouse gases, tannins are regarded as an important alternative in mitigating carbon dioxide (CO_2_), as well as methane (CH_4_) [63]. The same authors found that the addition of tannins into Nellore bulls’ urine had an effect on the reduction in CH_4._ OS van Cleef et al. [64] also concluded that the inclusion of highly taniniferous plants effectively mitigated the emission of CH_4_ in beef steers’ excreta.

## 8. Health Benefits of *Viscum* Species in Livestock

In many parts of the world, *Viscum* spp. have been consumed for a long period of time as an herbal tea and as a supplement to health care [8]. Furthermore, *Viscum* spp. have been used to improve livestock health, or simply as forage, when feedstuffs are limited due to drought [10,67]. Previous studies have highlighted that *Viscum* spp. have immunomodulatory, proapoptotic, and antimicrobial properties [8,68]. According to Ishiwu et al. [67], in Nigeria, rural farmers give the leaves of *Viscum* spp. to goats that have newly given birth, even though they do not, in reality, know of their health benefits. Moreover, Ohikhena et al. [61] highlighted that *Viscum* spp. are used to treat vision weakness and for promoting muscular relaxation prior to delivery. Drury [69] also highlighted the use of decoctions from *Viscum* spp. berries in cows to promote the expulsion of the afterbirth and to stop bleeding. In Nigeria, *Viscum* spp. are used to treat bacterial infections, skin conditions, diarrhea, diabetes, and prostate cancer in livestock [70]. It was reported that salmonellae in sheep rumen fluid were inhibited by diets containing *V. verrucosum* [71]. Further, Madibela and Jansen [72] highlighted that tanniferous species such as *Viscum* spp. can reduce the fecal egg count in ruminants. Apart from ruminants, Korean mistletoe enhanced lymphocytes and reduced *Salmonella* spp. of ceca in broiler hens [73]. Furthermore, *Viscum* spp. are used to treat infertility, epilepsy, rheumatism, and menopausal syndrome in humans [74].

## 9. The Use of *Viscum* Species in Ruminant Diets

The *Viscum* species are used as feed in many semi-arid and arid regions around the world, particularly during the dry season. *Viscum* spp. are rich sources of proteins and minerals, even though they contain antinutritional factors. Several studies have found that true mistletoes, when combined with other feed sources, can help reduce ruminant forage crop requirements in dryland areas. According to Jibril et al. [33], *V. album* can substitute sorghum stover by up to 50% in rams’ diets, without negatively affecting the growth performance. Similarly, Abubakar et al. [9] came to the conclusion that Red Sokoto Bucks can consume mistletoe leaf meal for up to 22.5% of their diet without any negative effects on the animals’ ability to produce. In an in vitro study, Ndagurwa and Dube [32] found that *V. verrucosum* had higher in vitro dry matter degradability, in vitro gas production, and in vitro metabolizable energy than *Acacia karroo*, thus making *Viscum* spp. a potential alternative browse for goats in semi-arid regions. Ramatsi et al. [2] also found that the in vitro dry matter degradability of *V. verrucosum* ranged from 510–517 g/kg at 72 h.

Apart from livestock ruminants, studies also show the positive effects of *Viscum* spp. in nonruminant livestock. It was reported that *V. album* improved the growth, meat, and carcasses of rabbits (*Oryctolagus cuniculus*), and can be added into rabbits’ diet at amounts of up to 15% [70]. Further, Ologhobo et al. [75] concluded that *V. album* had no effect on the growth performance, biochemical profile, and carcass characteristics of broilers.

## 10. Conclusions

*Viscum* spp. have the potential to serve as a substitute source of feed for ruminant animals due to their nutritional makeup, medicinal properties, and livestock acceptance. The utilization strategy will be of paramount importance and will be determined through establishing the correct mistletoe inclusion level in relation to low quality roughages. Even though the species does have detrimental impacts, it is advised that mistletoe control management in rangelands be conducted with caution. Moreover, the efforts taken to regulate it should not result in the plant’s total eradication due to the benefits it provides in terms of fodder, medicine, and in other areas. Future research may be required to assess the livestock preference of each species when present in different hosts. Again, there is a need to assess the palatability index of different *Viscum* species.

## Figures and Tables

**Figure 1 animals-12-02569-f001:**
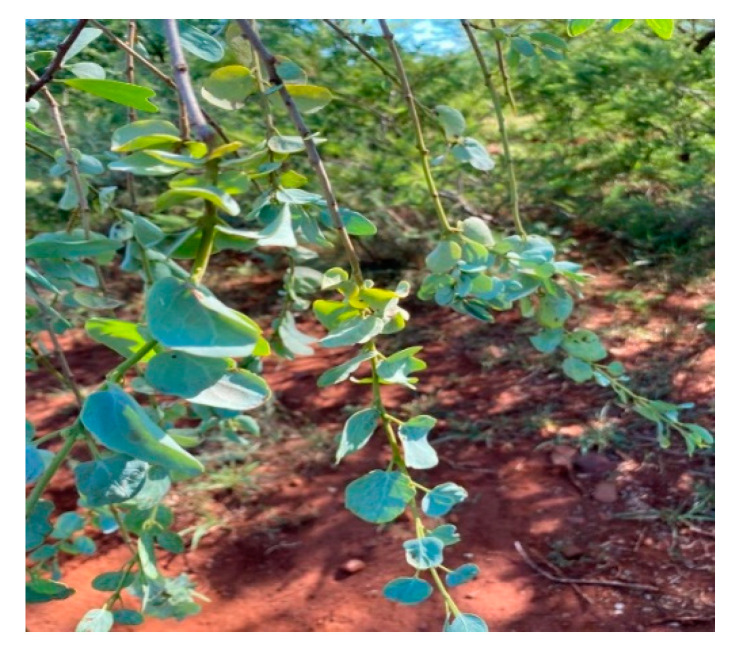
*Viscum rotundifolium* in Limpopo Province, photo taken by KE Ravhuhali.

**Figure 2 animals-12-02569-f002:**
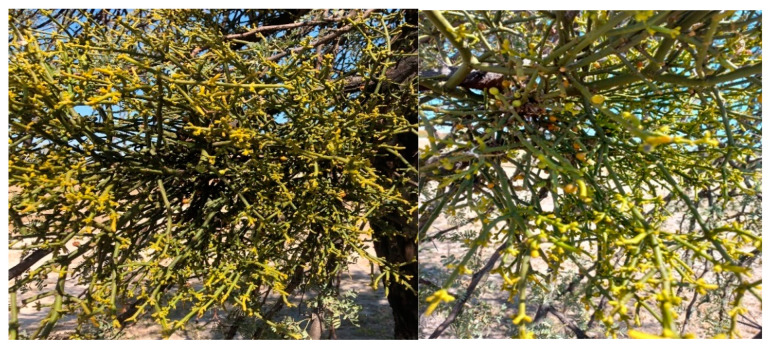
*Viscum verrucosum* Harv. in North West Province, photos taken by O Hawu.

**Table 1 animals-12-02569-t001:** *Viscum* species and their distribution.

	Distribution	References
*Viscum* spp. with leaves		
*V. articulatum*	Asia, Australia	[13]
*V. album*	Asia, Europe and Nepal	[14]
*V. cruciatum*	Asia, Africa and Europe	[15,16,17]
*V. rotundifolium*	Africa	[5]
*Viscum* spp. without leaves		
*V. angulatum*	Asia	[18]
*V* *. combreticola*	Africa	[19]
*V* *. anceps*	Africa	[20]
*V. songimveloensis*	Africa	[7]
*V. verrucosum* Harv.	Africa	[21]

**Table 2 animals-12-02569-t002:** Chemical composition (g/kg DM) of *Viscum* species.

Species	DM (g/kg)	CP	EE	NDF	ADF	ADL	References
*V. album*	960	150	80	339	202		[33,36]
*V. verrucosum*	912	121		276	244	75	[2,32]
*V. rontudifolium*		163			241	121	[37,38]

DM: dry matter, CP: crude protein, EE: ether extract, NDF: neutral detergent fiber, ADF: acid detergent fiber, and ADL: acid detergent lignin.

**Table 4 animals-12-02569-t004:** Antinutritional factors content (g/kg DM) in *Viscum* species.

Species	Tannins	Saponins	Phenolic	Oxalate	Phytates	Flavonoids	References
*V. album*	99	33		158	227		[65]
*V. rontudifolium*	7.3		28.3			2.4	[66]

## Data Availability

The data presented in this study are available on request from the corresponding author.

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
