# Peer review of "Utilization of the Viscum Species for Diet and Medicinal Purposes in Ruminants: A Review"

_animals, 2022, doi:10.3390/ani12192569_

Round 1

Reviewer 1 Report

Excellent work but if it requires some modifications: below I issue my observations from the review article "Utilization of Viscum (mistletoe) species as a diet and medicinal purposes for ruminants"

1.- Mistletoe is a very broad term that can include other species of the Loranthaceae genus, it can be limited to writing “True mistletoe” or “Viscum spp.” do not put Mistletoe spp.

2.- Line 42: missing dot after the V. Ex: V. songimveloensis

3.- Line 89. Correct “mistletoes contain crude protein (CP) content”

4.- Liena 90. Be specific that ruminant animals require 80 g/mg MS, that is, sheep, cattle, goats

5.- Line 99. What fiber does it refer to? The digestibility of a forage does not only depend on the fiber content. What other factors influence the digestibility of the forage?

6.- Line 105: better describe the role of minerals from parasitic plants in animal production.

7.- Line 146. add a section on the effect of secondary compounds such as tannins on the emission of greenhouse gases.

8.- In section 8, place another subtitle that includes the effects of the inclusion of mistletoe in other animal species in addition to ruminants.

9.- The subheading 9 Negative impact and control of mistletoe can be moved to the beginning of the review article to emphasize the importance of the use of these parasitic plants in livestock feed

Author Response

We would like to thank the Editors and reviewers for taking time to go through our article and providing constructive comments. We have gone through all the comments and our carefully considered responses are appended below. We hope that our responses are satisfactory, however we stand ready to make further changes should these be required.

Reviewer 1

Comment

Responses

Excellent work but if it requires some modifications: below I issue my observations from the review article "Utilization of Viscum (mistletoe) species as a diet and medicinal purposes for ruminants"

Thank you, for the positive appraisal of our work, we have now revised the copy.

1.- Mistletoe is a very broad term that can include other species of the Loranthaceae genus, it can be limited to writing “True mistletoe” or “Viscum spp.” do not put Mistletoe spp.

Thanks, we have now replaced mistletoe with Viscum spp. or true mistletoe

2.- Line 42: missing dot after the V. Ex: V. songimveloensis

Thanks for noting, corrected

3.- Line 89. Correct “mistletoes contain crude protein (CP) content”

Thanks for noting, corrected

4.- Liena 90. Be specific that ruminant animals require 80 g/mg MS, that is, sheep, cattle, goats

We have now specified the ruminant species 

5.- Line 99. What fiber does it refer to? The digestibility of a forage does not only depend on the fiber content. What other factors influence the digestibility of the forage?

We have now specified the fibres in the statement.

6.- Line 105: better describe the role of minerals from parasitic plants in animal production.

Thanks for the concern, the paragraph has covered parasitic plant such as Viscum species as the source of minerals for ruminants. We hope that the paragraph is satisfactory, however, we stand ready to make further changes should these be required.

7.- Line 146. add a section on the effect of secondary compounds such as tannins on the emission of greenhouse gases.

Thanks for your comment, we have now added statements regarding tannins on emission greenhouse gases.

8.- In section 8, place another subtitle that includes the effects of the inclusion of mistletoe in other animal species in addition to ruminants.

Thanks for your comment, we already highlighted the effect on rabbits, however, we have now added effect of Viscum album in broilers. As highlighted in Section 9 of the revised copy.

9.- The subheading 9 Negative impact and control of mistletoe can be moved to the beginning of the review article to emphasize the importance of the use of these parasitic plants in livestock feed

Thanks, we have now moved it (Section 4) before crude protein and fibre fraction of Viscum species

Reviewer 2 Report

Dear Authors,

Thank you for submitting this interesting review that considers mistletoe in the diets of animals. This review brings up some interesting points regarding the animal diets and the role of mistletoe in trees. There is some clear application in the management of livestock.

At current however, there seem to be some large revisions required in the manuscript to ensure the work is scientifically robust. I have attached the PDF version of the manuscript with specific comments. Additionally, please consider the following points: 

1. Nutritional values. Please provide a measure of variation for your nutrient values. Please also consider carefully the actual requirements of animals for minerals - as there is such thing as oversupplementation and undersupplementation.

2. Wording. There are some slightly confused sentences. As such, the work would benefit from a full proof read.

3. Structure. Consider the benefits and limitations of mistletoe presence in more detail. What should farmers do if they identify mistletoe in their areas? Provide some future directions for study in your work. 

Author Response

We would like to thank the Editors and reviewers for taking time to go through our article and providing constructive comments. We have gone through all the comments and our carefully considered responses are appended below. We hope that our responses are satisfactory, however we stand ready to make further changes should these be required.

Reviewer 2

Dear Authors,

Thank you for submitting this interesting review that considers mistletoe in the diets of animals. This review brings up some interesting points regarding the animal diets and the role of mistletoe in trees. There is some clear application in the management of livestock.

At current however, there seem to be some large revisions required in the manuscript to ensure the work is scientifically robust. I have attached the PDF version of the manuscript with specific comments. Additionally, please consider the following points: 

1. Nutritional values. Please provide a measure of variation for your nutrient values. Please also consider carefully the actual requirements of animals for minerals - as there is such thing as over supplementation and under supplementation.

2. Wording. There are some slightly confused sentences. As such, the work would benefit from a full proof read.

3. Structure. Consider the benefits and limitations of mistletoe presence in more detail. What should farmers do if they identify mistletoe in their areas? Provide some future directions for study in your work

Thanks for the positive appraisal of our work, we have now revised the copy.

Even though most of the sections have covered both limitations and benefits of the plant species, the following sections were much deeper on the benefits (Section 8 and 9) and limitations (Section 4). We have also conclude by going against the totally eradication of the species. Now we have added the omitted part in conclusion “future research”

Viscum spp.

Thanks, added

include scientific name here

Thanks, added

include common name for genus on first mention too.

There is no common name for “Vachellia” species

make sure this point is cited

References added

the term relished is quite subjective

We have changed to preferred

Is this all known species? If not, try to include all known species.

Though these may not be all known Viscum species, the one mentioned here were mostly found to be dominant in the literature.

These are useful. For identification purposes, please include images of the other species mentioned in the table above.

Thanks for the suggestion, the included pictures are for some of Viscum species mentioned in this review, we cannot assess all the pictures since some of the species are not available in our own area

Check whether this is the correct term on this occasion.

Thanks, we have now removed the term

Extract

Thanks for correcting

Not necessarily - it isn't because other food is expensive, but rather that the mistletoe is nutritionally suitable.

Thanks for correcting, we have now revised the statement.

Is there much variance in values? If so, provide a measure of such variance

Thanks for the input, however, we do not get what the reviewer might be looking for, the values in this table were extracted from different sources with some being converted into g/kg DM for consistence and uniformity. 

Again, could you report any variance in values?

Thanks for the input, however, we do not get what the reviewer might be looking for, the values in this table are average means as reported by the literature and the same adoption was used by several authors in their  published work. Eg. Ravhuhali et al. 2021, Sipango et al. 2022, Ntalo et al. 2022.

This is an interesting discussion but at current the content is too simple. What are suitable levels of Zinc, Calcium, etc in the diets of ruminants? This is important to consider as excess provision of some minerals results in toxicity.

Thanks for the comment, we have now included some of the mineral values as per certain specific animal requirement and the NRC citation was provided.

Mineral

Noted

Is there any information on the other species? Again, any variation between samples or studies?

There is limited information on the mineral content of Viscum species, hence we opted to list few species. This warrant the need for research (screening) on the assessment of mineral content in various Viscum species. 

Please check wording as this doesn't make sense.

The sentence has been modified to “There is an increasing interest in studying the bioactivity and anti-nutritional factors (ANFs) (phytochemicals) of Viscum spp.”

include scientific name here

Thanks, included

check formatting of year

Thanks, corrected

no need to write 'and' or 2021 here

Thank you for noting, it has been corrected

Italicise the journal name

Thanks corrected

Italicise the scientific name

Thanks corrected

italics?

Thanks, corrected

Reviewer 3 Report

Manuscript by Onke Hawu et al describing the possibility of the use of different species of mistletoe in diet and for medical purposes for ruminants is, in my opinion interesting and it deserves to be published in Animals. Mansucript conteins all the details and new aspects and in a very intersting way reflects the present state of knowledge. The literature, on which is based presented article is suffciently critical, current and internationally evaluated.

The size of the article is appropriate to the contents. and text is presented in a manner that scientists in other disciplines will understand. The text is presented and arranged clearly and concisely and abstract appropriately covers the content of the article. Key words are suitable so the article could be found in the current registers and indexes. The title appropriately reflects the content of the article. 

In general article is interesting and deserves to be published. 

Author Response

We would like to thank the Editors and reviewers for taking time to go through our article and we appreciate the positive input.

Reviewer 3

Manuscript by Onke Hawu et al describing the possibility of the use of different species of mistletoe in diet and for medical purposes for ruminants is, in my opinion interesting and it deserves to be published in Animals. Mansucript conteins all the details and new aspects and in a very intersting way reflects the present state of knowledge. The literature, on which is based presented article is suffciently critical, current and internationally evaluated.

The size of the article is appropriate to the contents. and text is presented in a manner that scientists in other disciplines will understand. The text is presented and arranged clearly and concisely and abstract appropriately covers the content of the article. Key words are suitable so the article could be found in the current registers and indexes. The title appropriately reflects the content of the article. 

In general article is interesting and deserves to be published. 

Thank you for positive appraisal of our manuscript review.

Round 2

Reviewer 2 Report

Dear Authors,

Thank you for providing a revised version of this paper. The paper is now in a  better position from a scientific standpoint.